

# Site selection by geese in a suburban landscape

Quentin J. Groom[1], Tim Adriaens[2], Claire Colsoulle[3], Pauline Delhez[4] and Iris Van der Beeten[1]

[1] Meise Botanic Garden, Meise, Vlaams Brabant, Belgium
[2] Research Institute for Nature and Forest (INBO), Brussels, Belgium
[3] Université catholique de Louvain, Louvain-la-Neuve, Belgium
[4] Université de Liège, Liège, Belgium

## ABSTRACT

**Background**. In European and North American cities geese are among the most common and most visible large herbivores. As such, their presence and behaviour often conflict with the desires of the human residents. Fouling, noise, aggression and health concerns are all cited as reasons that there are "*too many*". Lethal control is often used for population management; however, this raises questions about whether this is a sustainable strategy to resolve the conflict between humans and geese when, paradoxically, it is humans that are responsible for creating the habitat and often providing the food and protection of geese at other times. We hypothesise that the landscaping of suburban parks can be improved to decrease its attractiveness to geese and to reduce the opportunity for conflict between geese and humans.

**Methods**. Using observations collected over five years from a botanic garden situated in suburban Belgium and data from the whole of Flanders in Belgium, we examined landscape features that attract geese. These included the presence of islands in lakes, the distance from water, barriers to level flight and the size of exploited areas. The birds studied were the tadornine goose *Alopochen aegyptiaca* (L. 1766) (Egyptian goose) and the anserine geese, *Branta canadensis* (L. 1758) (Canada goose), *Anser anser* (L. 1758) (greylag goose) and *Branta leucopsis* (Bechstein, 1803) (barnacle goose). Landscape modification is a known method for altering goose behaviour, but there is little information on the power of such methods with which to inform managers and planners.

**Results**. Our results demonstrate that lakes with islands attract more than twice as many anserine geese than lakes without islands, but make little difference to Egyptian geese. Furthermore, flight barriers between grazing areas and lakes are an effective deterrent to geese using an area for feeding. Keeping grazing areas small and surrounded by trees reduces their attractiveness to geese.

**Conclusion**. The results suggest that landscape design can be used successfully to reduce the number of geese and their conflict with humans. However, this approach has its limitations and would require humans to compromise on what they expect from their landscaped parks, such as open vistas, lakes, islands and closely cropped lawns.

Corresponding author
Quentin J. Groom,
quentin.groom@plantentuinmeise.be

## INTRODUCTION

In Europe and North America, wild and feral geese frequently inhabit lakes and their surrounding parks in urban and suburban areas. These parks are appreciated by people for their recreational and aesthetic value. However, this often brings geese in conflict with people (*Conover & Chasko, 1985*; *Hughes, Kirby & Rowcliffe, 1999*; *Smith, Craven & Curtis, 1999*; *Fox, 2019*). While people often enjoy seeing small numbers of geese, when there are large flocks the soil becomes fouled and people are intimidated by the geese's threatening behaviour (*Miller et al., 2001*). Geese are also known to exert pressure on small water bodies such as ponds, reducing water quality through eutrophication (*Allan, Kirby & Feare, 1995*; *Gosser, Conover & Messmer, 1997*; *Smith, Craven & Curtis, 1999*; *Kumschick & Nentwig, 2010*). They have also been suggested to be a disease risk, though the evidence is circumstantial and other domestic and wild animals pose a greater known risk (*Fleming & Fraser, 2001*; *Clark, 2003*; *Bönner et al., 2004*). Throughout Europe and the western Palearctic, native as well as non-native geese are increasing in numbers and distribution (*Allan, Kirby & Feare, 1995*; *Fox et al., 2010*). Several populations have developed a resident component and their year-round presence increases human-wildlife conflicts and impacts on biodiversity (*Buij et al., 2017*). A variety of strategies are needed to reduce these impacts (*Austin et al., 2007*; *Gyimesi & Lensink, 2012*).

In Europe, from the 18th century onwards, it has been traditional to create landscaped parks reflecting an idealised vision of the countryside. Lakes with islands, open vistas, lawns and patches of woodland are typical (*Turner, 1985*). Lake-side vegetation and lawns are cut regularly and the canopies of trees are kept high to ensure unimpeded views. For those goose species that are habituated to the presence of people, such landscapes are very suitable, they have abundant grazing; proximity to water and islands for undisturbed nesting sites. In addition, people often provide supplementary feeding.

In north-western Europe, four species of "geese" are the main inhabitants of urban and suburban parks: non-native Egyptian geese (*Alopochen aegyptiaca*), Canada geese (*Branta canadensis*), mixed populations of wild and feral greylag geese (*Anser anser*) and barnacle geese (*Branta leucopsis*) (*Fox et al., 2010*; *Huysentruyt et al., 2019*). All are members of the family Anatidae, but Egyptian geese are members of the subfamily Tadorninae, which are referred to as tadornine geese, whereas the others are members of subfamily Anserinae, which are referred to as anserine geese (*Livezey, 1996*). Egyptian geese are similar in several aspects to anserine geese, such as their large size, long neck and feeding behaviour, but they do differ in other important aspects. Anserine geese, such as Canada geese, barnacle geese, greylag geese and their hybrids, usually nest on the ground close to bodies of water and are also likely to form large flocks (*Adriaens et al., 2019*). Egyptian geese are also water birds, but their biology shows many characteristics of a duck, including larger clutch sizes. Although they nest on the ground, their nest site selection is highly variable and they also nest in large tree holes, on buildings, on top of willow trees or in nest boxes (*Gyimesi & Lensink, 2012*; *Huysentruyt et al., 2020*). They also differ in their social behaviour. Paired Egyptian geese defend territories near their nest site before and during nesting. Large flocks of Egyptian geese only occur after breeding during moulting (*Gyimesi & Lensink, 2010*).

The site selection criteria of geese are important, because their sites can bring them into conflict with people. The proximity of water, food and breeding sites are relevant to goose site selection, but there are likely to be additional influences. These habitat features may be related to predator avoidance (*Conover & Kania, 1991*), accessibility of feeding grounds for adults and families with chicks, nutritional quality of feed (*Owen, Nugent & Davies, 1977*; *Fox & Kahlert, 2005*), sward length (*Hassall, Riddington & Helden, 2001*; *Feige et al., 2008*; *Conover, 1991*; *Van Gils et al., 2009*; *Huysentruyt & Casaer, 2010*) and competition with other grazers such as other geese, livestock and rabbits (*Van der Wal, Kunst & Drent, 1998*). Given this, it may be possible to identify management strategies and landscape features that alter the site selection of geese and these might be used to control the geese in such a way to reduce conflict between geese and people (*Conover, 1992*; *Owen, 1975*).

Culling is often used to reduce the impact of geese (*Reyns et al., 2018*), but several other strategies have been used to discourage and redistribute geese, including birds scarers and chemical antifeedants (*Conover, 1985*), fencing of feeding grounds or landscape modification, including altered mowing regimes or landscaping solutions (*Cooper, 1998*; *Van Daele et al., 2012*). In the context of a landscaped park with large numbers of visitors, culling risks losing public support for a public garden and bird scaring might disturb people too. At the same time, a botanic garden needs to consider the impact of grazing and fouling on plantings, lawns and vegetation, without losing the recreational opportunities for wildlife watching provided by the presence of these attractive birds. Therefore, habitat modification is considered as a cost effective, sustainable solution to reduce numbers of geese on sites and to mitigate the impact (*Conover, 1992*). Previous studies on site occupancy of geese have concentrated on wild geese in more or less rural settings. These studies have concentrated on ways to discourage geese from feeding on crop plants (e.g., *Olsson, Gunnarsson & Elmberg, 2017*; *Si et al., 2011*). In the case of Canada geese, most studies have occurred in North America (e.g., *Conover, 1992*).

The aim of this study is to quantify the site selection of the different species of geese within Meise Botanic Garden (Belgium) and create models to predict their behaviour based upon the landscape of the Garden. These models can then be used to suggest strategies to reduce conflict between the geese and the visitors to the Garden without losing the opportunities they represent for wildlife watching.

## MATERIALS & METHODS

Most of the research was conducted at Meise Botanic Garden (Flanders, Belgium), situated just north of Brussels, Belgium (50°55′42.4″N 4°19′37.6″E). The exception was the study on the effect of islands and those data are described below. The 92 ha Garden is a landscaped park like many such parks in northern and western Europe. It has extensive lawns, woodlands, two large lakes and one small one (Fig. 1). The Garden is subdivided into different numbered areas, divided by paths, which join various historic buildings and greenhouses with formal gardens, with approximately half the area covered by woodland. Most of the grassland is mown between two and four times a month during the growing season, though small areas are maintained as wildflower meadows and are cut once or twice

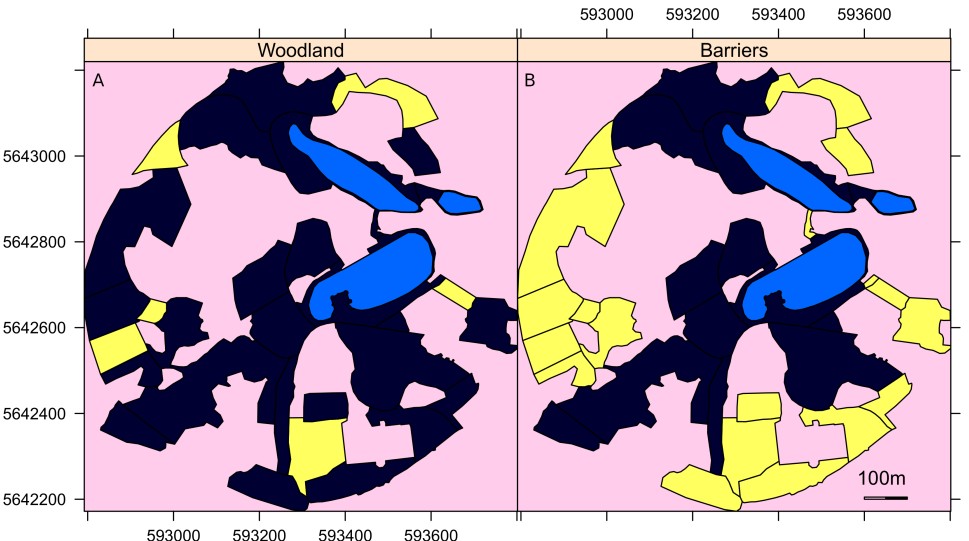

**Figure 1 A map of the surveyed areas of the Garden.** Maps of the Botanic Garden Meise where yellow indicates (A) the areas of woodland and (B) those areas largely surrounded by tall trees that act as barriers to direct flight of the geese out of that area. Light blue areas are lakes and pink areas were not surveyed. The unsurveyed areas are either covered by woodland, buildings or greenhouses. The axis are the UTM coordinates, zone 31U.

a year. All geese in the Garden are considered either non-native or feral. All species breed in the Garden, though the breeding of Canada geese is, in part, controlled by egg-shaking. The birds using the Garden are part of a larger population of geese that inhabit the greater Brussels area, and birds move in and out of the Garden to the many other lakes and waterways in the neighbourhood. None of these populations are truly migratory, except for local movements (*Anselin & Cooleman, 2007*). Canada goose is under management in the region and flocks of geese are regularly moult captured on water bodies in neighbouring municipalities since 2010 (*Reyns et al., 2018*). The Garden is in almost constant use by geese except for on the rare occasions when the lakes freeze over for long periods in the winter. Geese feed on all the lawns and grasslands within the Garden, but the extent to which these areas are used varies considerably from area to area and from species to species.

### The preference for grazing areas

The usage by geese of the different areas of the Botanic Garden was assessed by fixed transect counts (*Groom, 2019a*; *Groom, 2019b*). A total of four routes around the Garden were used, each route took approximately 40 min to walk and was always walked in a anticlockwise direction. Almost all of the grassland areas of the Garden were counted on at least two of these routes, woodland sectors were only counted when they were on the route between grassland areas.

Transect counts were conducted between 12pm and 2pm Central European Time. Geese were counted on an average of 2.7 days per week spread throughout the survey period that lasted nearly 6 years, between 11 Oct 2011 and 10 July 2017. Counts were conducted only on Monday to Friday at the convenience of the surveyors, but irrespective of weather
conditions. The only consistent period of the year when surveying was not conducted was between 25th December and 1st January. On a few occasions, two routes were walked simultaneously to give an approximate number for the total number of geese in the Garden for that day. Routes 1 and 2 gave the best coverage for all the main areas used by geese in the Garden. On other days routes 1 to 4 were chosen at random (*Haahr, 2019*). All the observation data are available on the Global Biodiversity Information Facility (*Groom, 2019c*).

It has been well argued, with good justification, that detectability is an important consideration in site occupancy modelling of animals (*Kéry & Schmidt, 2008*). Nevertheless, geese are large, noisy and bold and easy to recognize apart from the occasional hybrid. The areas where they feed in the Garden are small and open. Therefore, counts of the geese are expected to be reliable. We have not considered detectability in our analysis as we have no reason to think that this would make a difference to the results.

In one year, four hybrids were observed, two between greylag and Canada geese and two between barnacle and Canada geese. Furthermore, many of the greylag geese were either escapes from captivity or hybrids with farmed birds. Nevertheless, such distinctions were not made during counting and hybrids were counted along with the species they consorted with.

Three landscape parameters were examined for their importance for geese in site selection: the size of the survey area, the distance from the site to the nearest lake and the presence of physical barriers preventing direct flight to the nearest lake. Details of each survey sector are available in *Groom (2019b)*. For the physical barriers, each area was evaluated as to whether it was surrounded by barriers, such as tall trees and buildings that prevented easy flight access either to or from the lakes to the sector (Fig. 1).

These data have several issues which need to be addressed in statistical models. These are seasonal variations in behaviour, temporal autocorrelation and potentially spatial autocorrelation. Various statistical modelling approaches were considered including generalized linear models, mixed effects models and time series models. However, although these techniques might be useful to extract other valuable information from these data, we determined that, for the questions we wanted to answer, we would fit linear models to the mean individual count per sector. By averaging site occupancy across time, we eliminate the issue of temporal autocorrelation. Model selection was achieved by stepwise simplification of the model as described in *Crawley (2012)*, using the step and lm functions of R (*Venables & Ripley, 2002*). Independent variables were the area of the sector; the closest distance from the sector to the nearest lake; whether the sector was woodland (1) or grassland (0) and the presence or absence of flight barriers out of the sector towards the lakes. The log of the mean individual count per sector was our dependent variable. Evaluation of our initial models using residuals versus leverage plots showed that the sectors containing lakes (13, 18 & 21) had a disproportionate influence on the models as judged by the Cook's Distance. This is not surprising as the behaviour of geese and their relation to these areas is very different to grassland areas they visit to graze. For this reason, the lake sectors of the Garden were excluded from our models. This reduced the number of sectors used for the model to 29, but no sector had a disproportionate influence on the models. Residuals verses fitted

Q-Q plots were used to test whether residuals were normally distributed. A scale-location plot was used to test for homoscedasticity, meaning that the variance of the residual is homogenous across the range of the model. R version 3.4.1 was used in all modelling and data manipulations.

## Edge effects between grassland and woodland

Where goose grazing lawns are bordered by woodland it is reasonable to expect an edge effect, whereby the difference in usage by geese at a woodland-lawn boundary is gradual rather than abrupt. These might be the result of decreased forage quality in the partial shade of trees, or perhaps the avoidance of areas that give cover to potential predators. The use by geese of different areas of lawn was estimated by the amount of droppings on the lawn. Geese defecate frequently and seemingly indiscriminately. Counting dropping is a well-known method for estimating relative intensity of goose grazing on areas of land (*Owen, 1971*; *Van Gils et al., 2009*). However, we found it difficult to distinguish individual defecation events, because the droppings tend to break apart as they are released. Therefore, we preferred to measure the total length of droppings in a unit area. We considered this measure more reliable than trying to count the number of defecation events.

The presence of edge effects was investigated with 10 m wide rectangular plots laid out on the lawns perpendicular to the woodland-lawn boundary. The first set of four plots were 12m long and were surveyed in July 2014. The second set were 15m long and surveyed in March and April 2015. These plots are detailed in Table S1. The sites for these plots were chosen because they were on sections of the Garden frequently used by all goose species; well separated from each other; were away from other trees and faced different directions. The plots were marked out using bamboo canes and a tape measure. Then either 20 or 30 randomly chosen 1 m² square quadrats were surveyed within the rectangular plot. The cumulative length of dropping in a quadrat was measured to the nearest centimetre with a ruler.

Analysis of these data was conducted using non-linear mixed effects models using the plot number as a random factor (*Crawley, 2012*). Calculations were performed using the 'nlme' package in R (*Pinheiro et al., 2016*). Two possible models were compared, a 3-parameter asymptotic exponential model and a 3-parameter logistic sigmoidal function, both with a positive intercept. Model comparisons were made using the Akaike information criterion. Models were conducted using distances perpendicular to the woodland - lawn boundary and for a control modelling was repeated with distances parallel to the woodland - lawn boundary.

## Summer goose count data to investigate the influence of islands

Only one of the three lakes in the Botanic Garden has an island and this is the primary nesting site of greylag, Canada and barnacle geese. Nevertheless, with only one island it is impossible to draw conclusions about the importance of islands on habitat choice. Therefore, we used a dataset of summering goose counts from Flanders, that includes the Botanic Garden (*Devisscher et al., 2016*). These annual counts of geese are collected by volunteers from bird working groups at set sites across Flanders, Belgium. They are

conducted simultaneously over one weekend in mid-July, to avoid double counts and when most species have completed their moult but are still found aggregated in larger groups on water bodies (*Adriaens et al., 2010*; *Adriaens et al., 2011*). These data are provided with the geographic centroid of the lake. The area of the lake was calculated by tracing it on a GIS system and the area of the lake included the area of any island in the lake. The presence of an island in the lake was determined from visual inspection of aerial photographs from Google Maps.

## RESULTS

### Do geese avoid proximity to trees?

During the study geese were rarely ever observed in woodland. Egyptian geese are occasionally found perched in trees where they nest, but rarely on the ground in woodland. It was hypothesised that this negative association with woodland would extend beyond the boundary between the woodland and lawns and be the cause of an edge effects on grazing.

Quantification of the length of geese droppings showed a clear edge effect at the border to woodland (Fig. 2). A shorter length of droppings was found close to the woodland, but this effect only extended 5–10 m from the boundary.

As a control modelling was also performed in parallel to the woodland boundary, but models either failed to converge or showed no directional trend.

### Which habitat features attract geese?

Here we model the site selection of geese based upon habitat features we suspect might be important to geese. The area of the sector, barriers to flight, presence of woodland and proximity to lakes all appear relevant from observations of geese and the literature cited in the introduction. The mean individual counts of geese in the different sectors of the Garden are mapped in Fig. 3. From these maps it is clear that all species had a high affinity to the sectors containing lakes, though there are clear differences between species. The greylag geese in particular are far more wide-ranging than other species notably in the large western sectors.

The models of sector usage were evaluated with various means. The Cook's distance was used to evaluate if particular sectors had an exaggerated influence on the model outcomes, but this does not appear to be the case (Fig. S1). Variograms of the residuals did not show evidence for spatial autocorrelation that was not accounted for in the model parameters (Figs. S2–S5). A plot of residuals versus fitted values indicates that there may be some non-linearity between the predictors and the abundance of geese, but this was not clear (Fig. S6). The Q–Q plot shows that the residuals were quite normally distributed for all models (Fig. S7). The Scale-Location plot showed that some heteroscedasticity was evident in all models, however we consider that only the model for *B. leucopsis* was so heteroscedastic that it might impact our interpretation of the results. Given that no real-world model will perfectly match our assumptions and some of the reasons for deviation from these assumptions are suggested in the discussion.

A summary of the minimum adequate models is given in Table 1. The simplest minimum adequate model selected was for *Anser anser*. Only the area of the sector and the presence

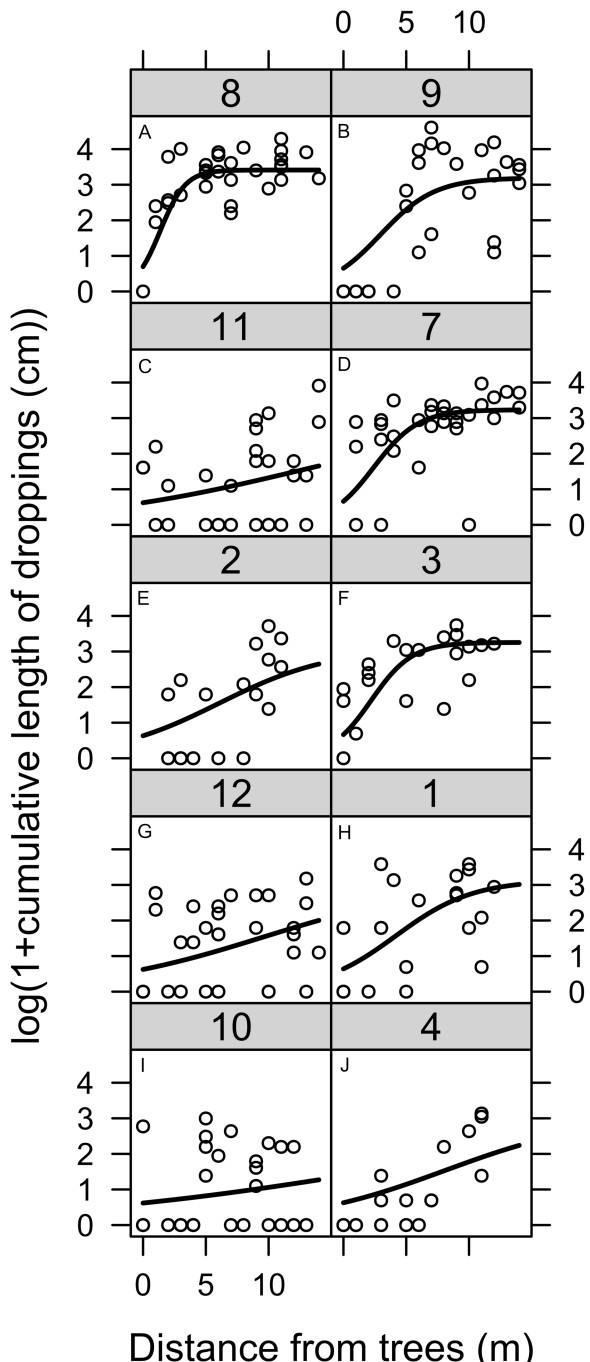

**Figure 2** **Geese land usage measured by the droppings deposited at varying distances from the boundary between woodland and lawn.** The total length of geese droppings deposited at varying distances from the boundary between woodland and lawn. Geese dropping were the sum length of all dropping from all species of geese. The numbers on each graph refer to the original plot number. See the Methods section for details of the model applied to the data.

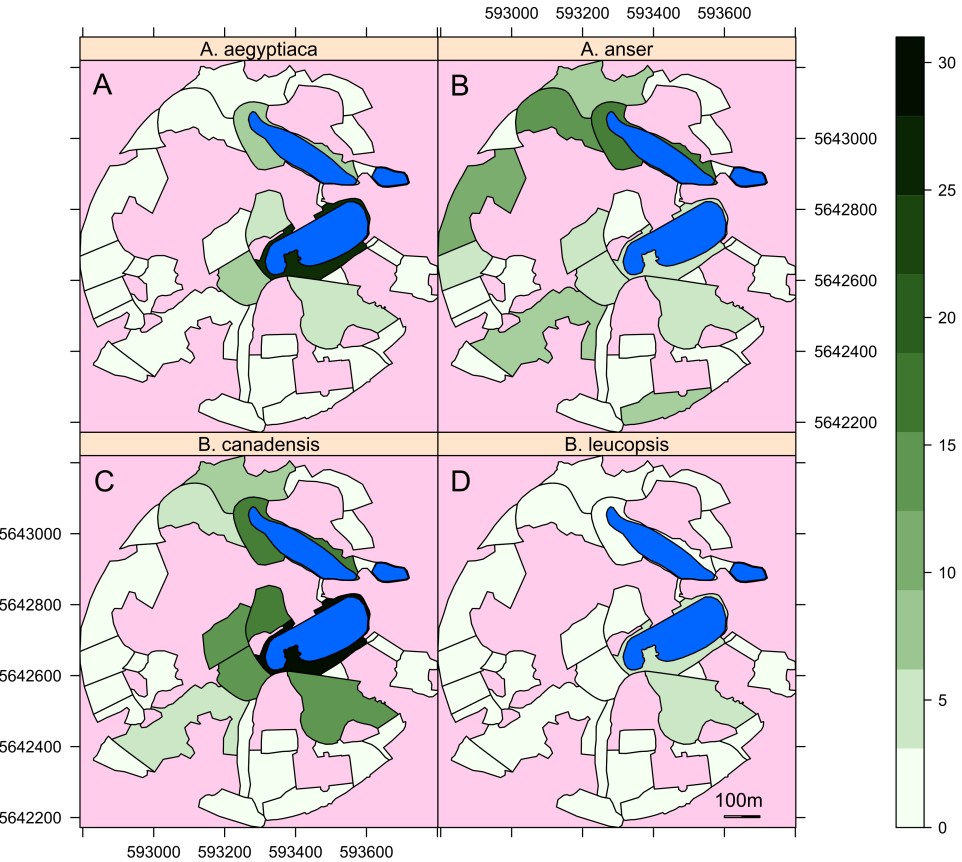

**Figure 3** **Maps of the area of the Garden used by the different species of geese.** Maps of the mean number of individuals of (A) *Alopochen aegyptiaca*, (B) *Anser anser*, (C) *Brata canadensis* and (D) *B. leucopsis* in the surveyed areas of the Botanic Garden. Lakes are in light blue, unsurveyed areas are in pink. The axis are the UTM coordinates, zone 31U.

of woodland were significantly correlated to their distribution in the Garden. Note, that these models do not include areas of the Garden containing a lake. For *B. canadensis* the area was also positively correlated with the number of geese, but not significantly in the model. However, in contrast to *Anser anser*, distance from a lake was a significant factor for *B. canadensis*, but also barriers to direct flight and their interacting term. For *Alopochen aegyptiaca*, area and barriers are significant as single factors, and they reoccur in interacting terms. Distance from the lake was not a significant term, but it did occur in an interaction term with area. In the case of *B. leucopsis*, area was a significant correlate, the other terms are more difficult to interpret, but both distance from a lake and the presence of barriers remained in the model due to their interactions and their interaction with area.

Goose abundance was negatively correlated with woodland for all except *B. leucopsis*, but this variable is not ideal as all those areas of woodland are also surrounded by trees as barriers to flight, So, there are no areas of woodland without barriers. Therefore, some of the variance stemming from the presence of woodland may be being accounted for in the barrier variable.

**Table 1  A summary of the minimum adequate models results for the distribution of geese in the Garden.** A '+' indicates a positive association of geese numbers with the independent variables and '−' indicates a negative association. The independent variables are the area of that sector of the garden, the distance from a lake, the presence of woodland on the garden sector and barriers to direct flight out of a sector. The number of asterisks indicate the degree of significance (*$p < 0.05$; **$p < 0.01$; ***$p < 0.001$). Details of the models are presented in Tables S2–S5.

| | *Alopochen aegyptiaca* | *Anser anser* | *Branta canadensis* | *Branta leucopsis* |
|---|---|---|---|---|
| Area | +* | +*** | + | +** |
| Distance from a lake | + | | −*** | + |
| Woodland | − | −** | − | |
| Barriers to direct flight | −* | | −*** | + |
| Area:distance | −* | | | −* |
| Area:barriers | +* | | | − |
| Distance:barriers | | | +*** | − |
| Area:distance:barriers | | | | +* |

Therefore, for all species the area of the sector was positively correlated with goose abundance and the area was part of the significant interactions included in the models for *Alopochen aegyptiaca* and *Branta leucopsis*. The distance from the lake remained in models for all species, except *Anser anser*. This is also evident in Fig. 3, where *Anser anser* can be seen to range more widely than other geese. All other predicted habitat determinants were included in one or more of the models.

For Canada and greylag geese there was a negative influence of barriers on site usage, particularly for Canada geese. In the case of Egyptian and barnacle geese, barriers were not a clear determinant of site selection, but did remain in minimum adequate models as interactions with distance and area.

## Do islands in lakes attract geese?

Lakes with islands attract more Canada, greylag and barnacle geese in the summer (Fig. 4). These results indicate that a lake without an island had 35%–60% fewer anserine geese than a lake of an equivalent size with an island ($p < .05$). However, islands made no difference to the number of Egyptian geese. All goose numbers showed a positive relationship with lake size ($p < 0.05$), although this is not significant in the case of barnacle geese.

## DISCUSSION

The modelling results, edge effects and impact of islands demonstrated the complicated relationship between habitat choice and the landscape for suburban geese (Figs. 1 and 2, Table 1). A casual observer could assume that there is a rather passive relationship between geese and their landscape, but as with any other animal, geese are clearly actively selecting and using particular landscapes and landscape features suited to their preferences.

Edge effects are relevant to the usage of geese on lawns because they reduce the active area of use for the geese. Our methodology did not distinguish whether there are species differences, however, the effect was so distinct that we speculate that all species are

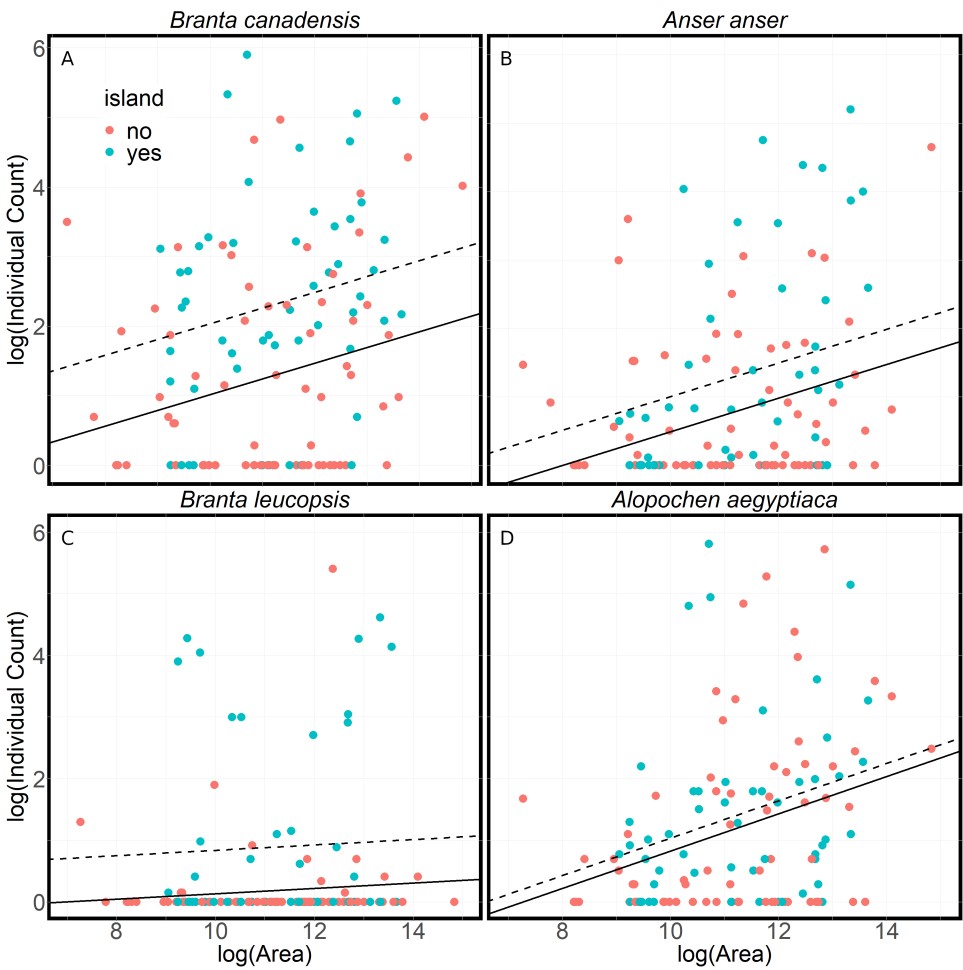

**Figure 4  A comparison of the numbers of geese found at lakes with or without islands.** A comparison of summer goose counts for lakes in Flanders compared to the lake area, either with islands (dashed line) or without islands (solid line). The lines are the results of linear models of the log of the average individual count on a lake and the log of the area of the lake. The models assume a constant relationship between average individual count of geese and the lake's area (A) *Branta canadensis* ($R^2 = .16, F(2, 119) = 10.98, p < .001$) (B) *Anser anser* ($R^2 = .12, F(2, 118) = 8.16, p < .001$) (C) *Branta leucopsis* ($R^2 = .09, F(2, 118) = 6.16, p < .01$) (D) *Alopochen aegyptiaca* ($R^2 = .10, F(2, 118) = 6.77, p < .01$). There is a significantly larger number of Canada ($t = 3.79, p < .001$), greylag ($t = 2.22, p < .05$) and barnacle geese ($t = 3.42, p < .001$) on lakes with islands. There is a significant positive relationship between the lake area and counts of Canada ($t = 2.58, p < .05$), greylag ($t = 3.30, p < .001$) and Egyptian geese ($t = 3.58, p < .001$).

influenced. While there may be many potential causes of an edge effect, such as predator avoidance and poorer grazing, an area of lawn less than 20 m in diameter is likely to be undesirable to geese. However, with increasing ratio of area to circumference means that the relevance of this effect will diminish with increasing area. In ornamental parks individual specimen trees might extend the influence of this edge effect.

Sector area was the most consistent predictor of goose abundance (Table 1). This was anticipated, as more space can contain more geese. Yet in addition to the edge effects there are reasons to expect a more sophisticated relationship between goose number and

area. Firstly, anserine geese are social species forming large flocks and they may only select areas with sufficient capacity to hold the whole flock. Secondly, if an area is surrounded by tall trees the flight angle needed to enter and leave it from the air becomes progressively steeper the smaller the area becomes. Mature trees stand 15–20 m tall, but average vertical and horizontal airspeeds of geese are approximately 0.5 m s$^{-1}$ and 16 m s$^{-1}$ respectively (*Hedenström & Alerstam, 1992*). Therefore, to enter and escape a small area surrounded by trees they must either considerably steepen their descent or climb rate, or circle while gaining or losing height. Both of these strategies would be more energetically expensive (*Norberg, 1996*). For these reasons, it is not surprising that the area of the sector also appears in interacting terms in the models with barriers. Barriers particularly restrict movement of geese when flight is not an option, such as, when raising young or moulting. However, the negative influence of barriers was scarcely significant for *Alopochen aegyptiaca*. This may be a result of their behaviour of nesting in tree holes. Though they do not inhabit densely forested areas, their preferred habitat is open grassland with some trees in proximity to freshwater (*Cramp et al., 1984*; *Carboneras, 1992*; *Gyimesi & Lensink, 2012*). They defend territories around nest sites and therefore must be in proximity to trees (*Sutherland & Allport, 1991*).

Distance from lakes was not as important to site selection as had been assumed, and the interactions with area and the presence of barriers suggests that the ease of access to grazing is more important to site selection than the linear distance. This perhaps indicates that careful usage of landscape features could guide geese to use particular feeding sites, irrespective of their distance from the lake.

The results show a strong preference of anserine geese for lakes with islands during the summer (Fig. 4). Islands are used by geese year-round, as they provide protection from disturbance where geese can rest and nest. The lack of a similar preference for Egyptian geese is consistent with the territorial breeding behaviour of Egyptian geese and their use of nest holes in trees. Although anserine geese prefer lakes with islands in the summer, the reasons are probably many and this preference may not be true in winter. Island breeders are presumably more protected from predators, particularly foxes (*Vulpes vulpes*) (*Wright & Giles, 1988*), stone marten (*Martes foina*), brown rat (*Rattus norvegicus*) and carrion crow (*Corvus corone*) (*Huysentruyt et al., 2020*). However, when breeding success on islands has been examined it is not always better than on the mainland (*Gosser & Conover, 1999*; *Petersen, 1990*). Other studies on the influence of islands on goose nest site selection vary. Fox et al. (1989) showed no influence for greylag goose, whereas others report an effect for Canada Goose (*Lokemoen & Woodward, 1992*; *Bromley & Hood, 2013*). *Huysentruyt et al. (2020)*, in their study of 200 breeding pairs of barnacle goose in Flanders, also note that barnacle goose mainly breeds on small islands in lakes and ponds in the region.

Based on the results of this study we suggest that landscape adaptations could indeed reduce the number of geese in suburban parks, which could be an alternative to lethal control and prevent conflict with people. Unfortunately, many of the landscape adaptations that would reduce the presence of geese are in opposition to popular landscape design features, such as ponds and lakes, islands, open vistas and extensive lawns. Other sorts of landscape and garden design with more enclosed and higher vegetation are more suitable

where geese are a problem. Woodlands, shrubberies, coppice, hedges, tall grass meadows, prairie planting, hard landscaping features, shallow water and moving-water features would all deter geese from using an area (*Allan, Kirby & Feare, 1995*; *Gosser, Conover & Messmer, 1997*; *Allan, 1999*; *Baxter, Hart & Hutton, 2010*).

If artificial islands were eliminated from suburban lakes it might be argued that native birds would also suffer from the lack of island breeding sites, however, islands in suburban parks are mostly unsuitable for island nesters of conservation concern, such as common terns (*Sterna hirundo*) which do breed well on artificial rafts in bigger lakes and lagoons (*Coccon et al., 2018*; *Dunlop, Blokpoel & Jarvie, 1991*). Islands could perhaps be made less attractive if they were connected to the mainland by constructing bridges or an isthmus. They can also be modified with banks that deter access from the water, rather than from the air. However, making feeding areas inaccessible is controversial as chicks can then starve (*Allan, 1999*). Modifications or removal of islands should however consider the trade-off with ongoing management. For example, when practicing egg shaking or egg oiling for fertility reduction, the success of this measure depends on sustained effort and a high percentage of treated nests (*Klok et al., 2010*; *Beston et al., 2016*). Hence, having all geese nest on the same island makes it easier to perform this management.

There is also a need to educate the public to the benefits of geese. In the Botanic Garden their selective grazing of grasses has created an exceptional species rich grassland that is unlikely to be maintained with mowing alone (*Ronse, 2011*). An adaptive management approach, whereby vegetation and goose numbers in the Garden are thoroughly monitored and objectives are clearly stipulated, could be a good way to learn more about the behaviour and impacts of geese.

## CONCLUSIONS

Landscape features have a powerful influence on the distribution of geese, though these influences differ between species. For example, we show that...

- Lakes with islands attract more than twice as many anserine geese
- Flight barriers between grazing areas and lakes deter geese
- Small grazing areas surrounded by trees reduces their attractiveness to geese
- Proximity of a lake is most important to Canada geese, and least to greylag geese

Landscape modifications cannot completely remove geese from a suburban landscape and an integrated management strategy may be necessary (*Allan, Kirby & Feare, 1995*). Retroactively modifying landscapes to reduce their attractiveness to geese is difficult, so designing landscapes for wildlife usage should be among the primary design criteria.

## ACKNOWLEDGEMENTS

The authors would like to thank Didier Vangeluwe of the Royal Belgian Institute of Natural Sciences and Danny Swaerts of the Meise Botanic Garden for their support and advice during the project.

### Funding

The authors received no funding for this work.

### Competing Interests

The authors declare there are no competing interests.

### Author Contributions

- Quentin J. Groom conceived and designed the experiments, performed the experiments, analyzed the data, prepared figures and/or tables, authored or reviewed drafts of the paper, and approved the final draft.
- Tim Adriaens analyzed the data, authored or reviewed drafts of the paper, and approved the final draft.
- Claire Colsoulle, Pauline Delhez and Iris Van der Beeten performed the experiments, authored or reviewed drafts of the paper, and approved the final draft.

### Data Availability

Devisscher S, Adriaens T, Brosens D, Huysentruyt F, Driessens G, Desmet P (2019). NBN - Zomerganzen - Summering geese management and population counts in Flanders, Belgium. Version 1.2. Research Institute for Nature and Forest (INBO). Sampling event dataset 10.15468/a5ubtp accessed via GBIF.org on 2019-04-21.

Pauline Delhez, & Quentin Groom. (2019). Detecting edge effects of geese grazing at the boundary of woodland and grassland [Data set]. Zenodo 10.5281/zenodo.2648174.

Groom Q. 2019a. A map describing routes used in goose monitoring at Meise Botanic Garden. Zenodo. DOI: 10.5281/zenodo.2529725.

Groom Q. 2019b. Goose monitoring data from the Meise Botanic Garden, Belgium [Data set]. Zenodo. DOI: 10.5281/zenodo.2529723

Groom Q. 2019c. Waterbirds of the Botanic Garden Meise. Version 3.6. Botanic Garden Meise. Occurrence dataset DOI: 10.15468/cgffyq accessed via GBIF.org on 2019-01-05.

### Supplemental Information

Supplemental information for this article can be found online at http://dx.doi.org/10.7717/peerj.9846#supplemental-information.

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
