# Peer review of "Site selection by geese in a suburban landscape"

_PeerJ, doi:10.7717/peerj.9846_

## Round 0.1 · original submission · Major Revisions

Dear author

We have received the reviews from our reviewers. As you can see Rev 1 is very critical and even recommends a rejection. As I personally see value in your work, I would like to give you a chance to thoroughly revise your ms. Make sure that you address the recommendations of the reviewers, as they will review your revision.

Kind regards
Michael Wink
Editor

·

Basic reporting

The manuscript needs many more references, for example, the discussion is over 1000 words long and only contains one reference. Please look into the current literature in high-impact journals for examples.
The manuscript needs to be much more focused and more rigorously supported by references. As it is, large parts (e.g. most of the introduction and discussion) are highly speculative.
Results section: Contains sections that belong into the other sections, e.g. lines 276 – 279 belong in Material & Methods, 289 – 293 are speculative and sound more like a discussion.
The results need a lot more proof. E.g. if you state something is “significant”, give at least the p-value. Don’t expect the reader to go to the supplementary tables to check every result.
Figures and tables: Axes are often unclear – what are you showing? Captions are missing! Large parts of the body of the text can probably moves into figure/table captions.
References: you need to support each statement you make. If you give references, please don’t dump them all together at the end of a long sentence (e.g. lines 105-106), but put each reference right after the statement it belongs too. Make it easy for the reader!
Discussion: Needs a lot more structure. Please start by summarizing your main findings briefly in the first paragraph. Then start each paragraph with a finding, which you then discuss. Go from the most important to the less important results, and keep it short and concise.

Experimental design

Looks good. Good field data.

Validity of the findings

Interesting results, but need to be made a lot clearer. They sound too speculative. You currently give no hard data with p-values, numbers of geese, etc in the results. You have so much good data from so much fieldwork, I am sure you can greatly improve this!

Additional comments

I think the manuscript could benefit from proofreading by a native speaker.
It may benefit from being rewritten into a short communication format, as the findings don’t justify such a huge body of text.

Reviewer 2 ·

Basic reporting

The study reports on goose distribution by counts in a park in Belgium considering very rough measures of landscape (Woodland, distance to water). Edge effects are investigated using dropping length measures. Although some sections are well written, there are considerable problems with this paper:

1. Please give this manuscript to a native speaker to correct the English. There are many mistakes. I.e. Instead of “geese” often “goose” would be more appropriate.

2. In a lot of sections, literature references are missing and one cannot distinguish between published information and opinion of the authors.

3. The article structure is only in parts professional. A table is missing in the main section of the paper with the results of the models. Raw data are shared and supplemental figures (however, the latter a not explained).

4. The introduction provides good background and information, but some of it is irrelevant for the study of site selection, it should be shortened by at least 50 %. Relevant prior literature is not always appropriately referenced. A hypothesis is not formulated as is practice in scientific research? What are the predictions?

5. The order of things studies should be the same in intro, meth, res and dis. It is otherwise really confusing.

6. The discussion is unacceptable. References are entirely missing. I would not accept such a discussion even in a student report. It needs to be rewritten, in a more comprehensive manner including references. There are plenty relevant papers. It should put the authors results into relation to other studies. It should only discuss things based on the initial hypotheses and results and free of personal opinion.

7. Figures are relevant to the content of the article, of sufficient resolution, and appropriately described and labelled.

8. As far as I can see all appropriate raw data have been made available in accordance with our Data Sharing policy.

9. Not at all self-contained with relevant results to hypotheses.

10. A table with the summary of the models should be presented in the main section of the paper, not only as supplement.

Experimental design

It is stated that research fills an identified knowledge gap, but this is not true globally as papers have been overlooked that exist on the subject. However, as sites vary, it would still be relevant to conduct another study on the subject.

Rigorous counts performed but questions remain as to collection of dropping data (see comments in pdf). Areas not randomly chosen, only areas included that are “liked” by the geese), whereby the observer influences data collection subjectively. The technical standard is rather easy: Counting. The research has been conducted in conformity with the prevailing ethical standards in the field. Not all methods have not been described with sufficient detail & information to replicate (also see comments in pdf).
In the count models no consideration is given to seasonal effects, i.e. during moult behaviour of geese changes and may not be the same as in the rest of the season.

Validity of the findings

It is interesting to see the detailed distribution maps of the geese, edge effects are different in different habitats, so the idea is valid although there are other studies which are more refined. It is however disappointing, that habitat is described in such a poor manner.
The data on which the conclusions are based are available.
The conclusions are not all connected to the original question investigated. Speculation is not always identified as such.

Additional comments

Although the question at hand is a relevant one, you fail to provide some basic information, references and the structure of the paper is not coherent. In the introduction you set out with a very global view, but then you only describe goose distribution in one park in relation to a few parameters. You fail to report the behaviour the geese were showing during the count (feeding, resting, walking) and just assume that geese on pastures use them for feeding, which is not true). The manuscript needs shortening. This is not too difficult when leaving out sections that are not relevant to site selection. The method is rather crude and does not consider habitat choice by individuals. Some methods do not follow standards used in the field (random selection of plots, stratifying according to personal observation, which could be seen as critical).

Annotated reviews are not available for download in order to protect the identity of reviewers who chose to remain anonymous.

·

Basic reporting

Thank you for reporting your results on the overlooked relationship between geese abundance and landscape, in an urban area.

While I welcome your paper and I believe it brings an important output, I have two main comments:

1. You have a strong focus on islands and tree barriers in the first sections on the paper, but you somehow loose this in the Discussion. Although you mention something about the edges of the islands and its accessibility, nothing is written on the dimensions of the islands, vegetation, type of soil and substrate. For the tree barriers, again I believe some more emphasis should be delivered. Which tree species would bring more benefit for reducing geese numbers. At which height from the ground should the canopy start? Which density do you recommend? Or these factors have no importance at all?

2. When counting the geese in the garden, did you also take into account human presence? You mention the counts were done on Monday and Friday. I guess Monday is not the busiest day in the garden, but I expect Friday you have quite a lot of visitors. It is possible that the admission / entry office keeps a record on the number of visitors. If related to the number of geese, you think there might be a correlation?

Finally, throughout the article the English is sometimes tricky. Several times I've got myself lost in long sentences, where the verb was sometimes missing. I've corrected some of these mistakes, but please take a thorough inspection of the whole text. And I absolutely do not understand why "Egyptian" and "Canada geese" start with capital, but "greylag" and "barnacle geese" do not have capital. You use this system in the entire manuscript and after a few sentences I stopped correcting it.

Please check the annotated manuscript for further comments and details.

Experimental design

Please refer to the 'basic reporting' section and the annotated manuscript. All my comments, questions and suggestions are there.

Validity of the findings

Please refer to the 'basic reporting' section and the annotated manuscript. All my comments, questions and suggestions are there.

Additional comments

Overall, I found your manuscript as an interesting piece of research and I would be happy to see it published as soon as possible.
Please address my comments and I look forward to approve the second version.

With best regards,
Liviu Parau

---

## Round 0.2 · Major Revisions

Dear authors:

Please take the recommendation of the reviewers more seriously, otherwise, we will have to reject your paper. It is far better to follow the advice of the experts instead of simply providing a rebuttal of their arguments.

Regards
Michael Wink
AE

·

Basic reporting

The manuscript has been much improved. Some parts are however still lacking references (please see my comments below) and some parts are confusing.

Experimental design

Good experimental design. Results part could be clearer to make the main results stand out. Discussion could be clearer especially in terms of pinpointing main findings and connecting them back to the aims.

Validity of the findings

no comment

Additional comments

The manuscript has been much improved from the previous version and the study is very interesting. I however still have some comments, mainly around lack of clarity and references:

Introduction
Lines 79 – 84 provide reference
84 – 86 does not lead anywhere – omit
98 which features?
117 considered by whom?

Methods
Line 143 change to ‘Canada geese are’
130 – 149 you change keep switching between the terms ‘Garden’ and ‘park’, which I find confusing. If you’re talking about the same area, please keep the terminology consistent.
178 – 179 Please connect this sentence better with the previous one, e.g. ‘Three landscape parameters were examined for their importance for geese in site selection: the size of the survey area, the distance from the site to the nearest lake and the presence of physical barriers preventing direct flight to the nearest lake.’
183 change to ‘….be addressed in statistical models. These are…’
204 – 206 reference missing, and please define the term ‘edge effect’.

Results
256 – 259 is not a result. Please move or omit.
269 – 274 Define terms ‘homoscedasticity’ and ‘heteroscedastic’ in Methods. Make sure that you don’t mix methodology with results.
Figure 1 and 3 As I already pointed out in the first round of revisions, please explain the numbering/labels in your figures more thoroughly. Readers which are not too familiar with maps like the ones you are using will not be able to follow.
Figure 4 Please move the statistics (p-values etc) from the caption to the results text. It makes it easier for the reader to follow the story you’re telling.
In general, as mentioned in my earlier review, please improve clarity. What are your main findings?

Discussion
310 – 314 As I pointed out during the first revision, please give a brief but precise summary of your main results at the start of the Discussion. You mentioned a complicated relationship. Tell the reader why? What are you basing that on? You state that ‘geese are clearly actively selecting’ particular landscapes. Tell us instead what brings you to this conclusion? E.g. ‘In this study, we found that a) geese avoided woodland areas, b)…………etc etc. At the moment, you are losing the reader in the results section and don’t give them a chance to get back on track in the discussion.
316 – 322 Please provide references. E.g. line 319 which potential causes? Line 320 why do you expect this effect to diminish?

·

Basic reporting

See general comments.

Experimental design

See general comments.

Validity of the findings

See general comments.

Additional comments

As in my first review, I mention again that the English should be improved. Long, unstructured phrases make the reading difficult. Many Oxford commas are missing.

Furthermore, throughout the text, you are inconsistent with abbreviations. For example, between lines 390 and 411, all scientific names are given in full, except for Branta. However, sometimes you used “B.” and sometimes “Branta”.

Another example, at line 410: you write: “A. anser can be seen to range more widely than other geese”. I changed this to “A. anser can be seen to have a wider range than any other geese”.
Line 419: “Lakes with islands house more Canada, greylag and barnacle geese in the summer”. Changed to “Lakes with islands attract more Canada, greylag and barnacle geese in the summer”
Line 495: “the negative influence of barriers was barely significant”. Changed to “hardly significant”.

Finally, I strongly believe that this is a very important knowledge piece and should be published. Personally, I never regarded landscape structure as having such a decisive influence on urban and suburban geese abundance. But after reading your work, I will definitely pay more attention to park architecture.

Thank you for your efforts in collecting, analyzing and publishing the data!

---

## Round 0.3 · accepted · Accept

Dear authors

Although one of the reviewers is not happy that you did not follow their advice, we will accept your ms.

Regards
M. Wink
AE

·

Basic reporting

The language and literature have been improved from the first version. Raw data are shared, with the exception of raw p-values (for example, p-values are shared in categories like <0.5, instead of actual numbers (e.g.'0.03') given). The results are relevant to the aims given.

Experimental design

As mentioned before, the authors have tackled a topic which is relevant to many urban areas and have put a lot of work into collecting field data and analysing the resulting data.

Validity of the findings

no comment

Additional comments

no comment